# Distributed Management Systems for Infocommunication Networks: A Model Based on TM Forum Frameworx

**Valery Mochalov \*, Natalia Bratchenko, Gennady Linets and Sergey Yakovlev**

Department of infocommunications, North Caucasus Federal University, Stavropol 355017, Russia;
nb20062@rambler.ru (N.B.); kbytw@mail.ru (G.L.); Yak0vlevSV@yandex.ru (S.Y.)
\* Correspondence: mochalov.valery2015@yandex.ru

**Abstract:** The existing management systems for networks and communication services do not fully meet the demands of users in next-generation infocommunication services that are dictated by the business processes of companies. Open digital architecture (ODA) is able to dramatically simplify and automate main business processes using the logic of distributed computing and management, which allows implementing services on a set of network nodes. The performance of a distributed operational management system depends on the quality of solving several tasks as follows: the distribution of program components among processor modules; the prioritization of business processes with parallel execution; the elimination of dead states and interlocks during execution; and the reduction of system cost to integrate separate components of business processes. The program components can be distributed among processor modules by an iterative algorithm that calculates the frequency of resource conflicts; this algorithm yields a rational distribution in a finite number of iterations. The interlocks of parallel business processes can be eliminated using the classic file sharing example with two processes and also the methodology of colored Petri nets. The system cost of integration processes in a distributed management system is reduced through partitioning the network into segments with several controllers that interact with each other and manage the network in a coordinated way. This paper develops a model of a distributed operational management system for next-generation infocommunication networks that assesses the efficiency of operational activities for a communication company.

**Keywords:** Open digital architecture; OSS/BSS; Frameworx; business process framework; distributed management system; program component; CPN Tools

## 1. Introduction

Unlimited growth of network traffic, processes of convergence of networks and network devices and the need for new business models determine the growing importance of new methods of reducing operating costs, making efficient use of network resources, and reducing risks in managing digital businesses. However, the existing management systems for networks and communication services [1–5] are able to render next-generation infocommunication services with the required level of flexibility, management and cost-saving only in part.

The Tele Management Forum (TM Forum) [6], an international non-commercial association of telecommunication companies and their suppliers, is directing its partners towards a successful implementation of digital companies. TM Forum experts proposed the idea of open digital architecture (ODA) [7], which is expected to replace traditional operation support systems/business support systems (OSSs/BSSs) as well as to simplify dramatically and automate main business processes (BPs). ODA

functionality is intended to execute business processes without any human interference using advanced technologies, including artificial intelligence (Figure 1). As its key principle, the ODA involves the component approach with the logic of microservices and open APIs (Application Programming Interfaces). The main design concepts of such architectures, and also the new virtualization approaches for the physical resources of next-generation infocommunication networks, were described in the ITU-T Recommendations, see the Y.3000–Y.3499 series [8]. More specifically, services are formed through the integration of program components represented by the elements of a distributed system; services are abstracted from network resources and their arrangement, thereby implementing network virtualization: multiple services–single network. The business processes implementing the main services of a communication company are the integration base for program components (PCs). In other words, management system design involves a set of interconnected PCs that are intended to perform separate partial tasks for BP execution. The resulting adaptive management system can be easily reconfigured to any BP. The set of PCs matches the Enhanced Telecom Operations Map (eTOM) library modules: all necessary actions for creating a new service are implemented by the PCs connected with each other through the corresponding business processes.

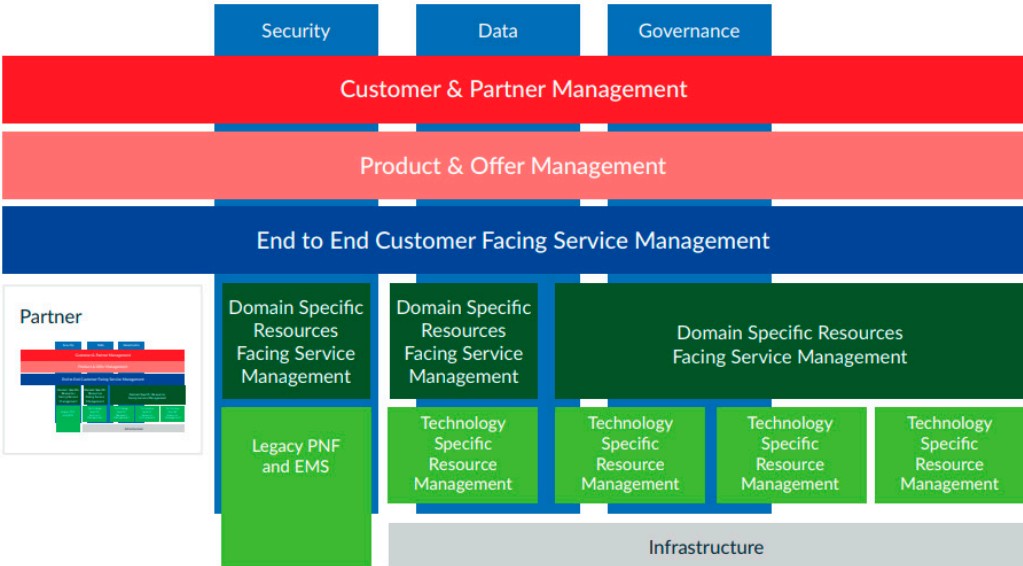

**Figure 1.** The Tele Management Forum TM Forum open digital architecture [7].

## 2. Results

Using the component approach, a model of distributed operational management systems for next-generation communication networks was developed that assessed the efficiency of operational activities of a communication company.

In particular, the results include the following.

- the formal design problem of distributed operational management systems for communication companies and the main approaches to this problem;
- an iterative convergent algorithm for distributing program components among program modules that implements a corresponding microservice or business component in the Frameworx description;
- a service schedule of requests with the colored Petri nets formalization that minimizes the number of delayed requests;
- an integration algorithm for program components that minimizes the system cost of their interaction.

All these results contribute to an efficient design of distributed operational management systems for next-generation telecommunication networks and also determine a flexible, software-defined and cost-saving architecture of management systems for infocommunication networks and services.

## 3. Methods

### 3.1. Frameworx TM Forum

The ODA project was implemented using the concept of Frameworx TM Forum, known earlier as the New Generation Operations Systems and Software (NGOSS). This concept defines a modern standardization approach for the business processes of a communication company [9]. Frameworx gives an exact description of the components of BPs in terms of their functions, associated information and other characteristics. Frameworx consists of the following frameworks (Figure 2).

1. The business process framework (formally the enhanced telecom operations map, or eTOM) describes the structure of the business processes of telecommunication companies.
2. The information framework (formally the shared information/data model, or SID) defines an approach to the description and usage of all data engaged in the business processes of a communication company.
3. The application framework (formally the telecom applications map, or TAM) describes the typical structure of the information framework components for communication companies.
4. The integration framework contains a set of standards that support the integration and interoperability between applications defined in the applications framework, with a basic element in the form of a standardized interface; a set of similar interfaces defines the service (API service).
5. Business metrics are a standardized model of business indicators that unites over a hundred of standard measurable indicators for assessing the different activities of an infocommunications supplier.
6. Best practice includes practical recommendations and case studies based on the experience of using Frameworx models in different activities of telecommunication companies.

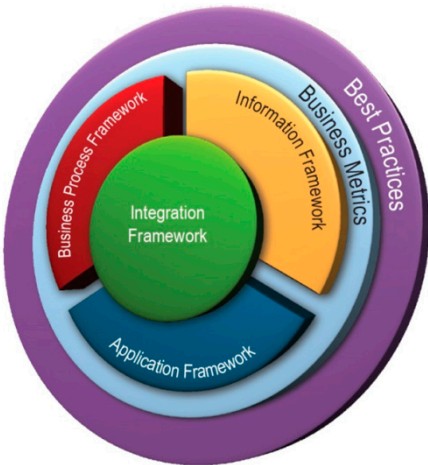

**Figure 2.** Structure of Frameworx TM Forum [9].

The hierarchical decomposition principle is widely used for the structural description of BPs in eTOM. Consider the decomposition procedure for BP 1.4.6.3 (Correct and Resolve Service Problem); there are four decomposition levels here, as is illustrated in Figure 3.

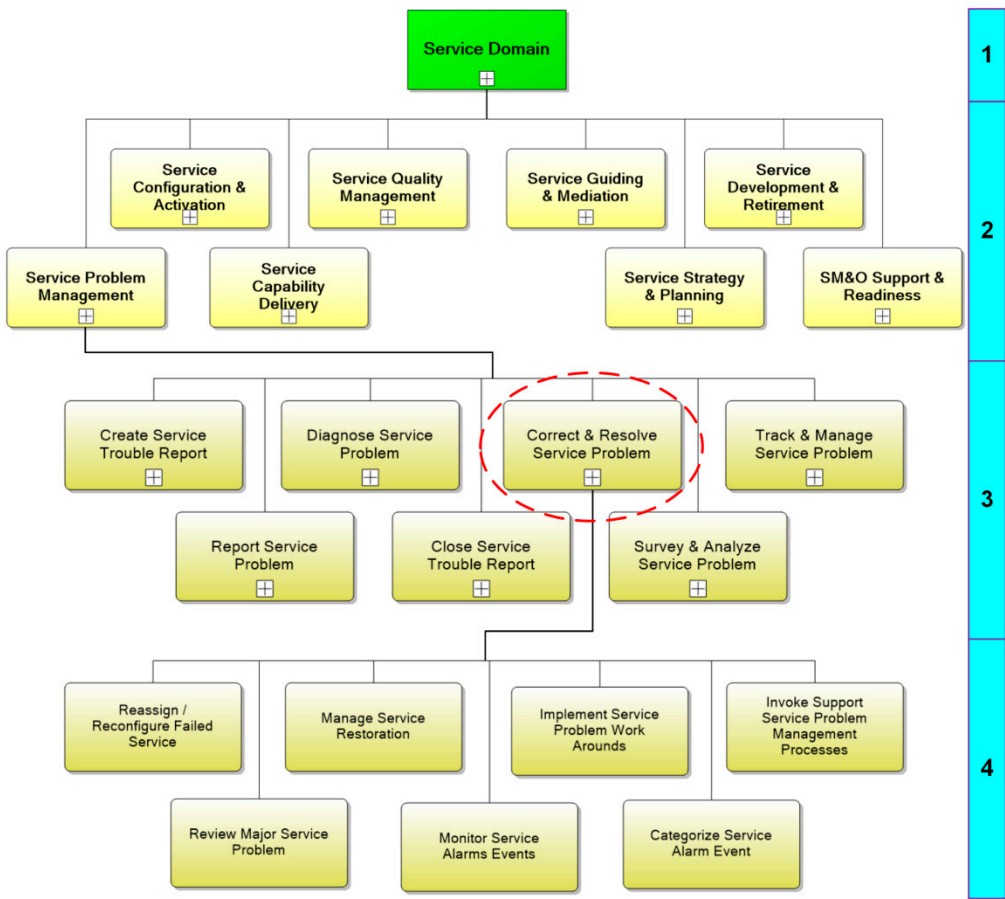

**Figure 3.** Decomposition of business process (BP) 1.4.6.3 (Correct and Resolve Service Problem) [9].

The first decomposition level of eTOM includes eight large blocks as follows:

- Market/Sales Management Domain;
- Product Management Domain;
- Customer Management Domain;
- Service Management Domain (is shown in Figure 3);
- Resource Management Domain;
- Engaged Party Domain;
- Enterprise Domain;
- Common Process Patterns Domain.

The second decomposition level separates the groups of processes that represent large stages of end-to-end business processes in eTOM. For example, block 1.4 (Service Management Domain) at the second level is divided into eight groups; see Figure 3. In particular, it includes block 1.4.6 (Service Problem Management).

The described levels are logical because the resulting specification does not yield a sequence of actions. The third and lower decomposition levels are physical, since their elements correspond to specific actions that can be combined in flows.

The processes of the third decomposition level can be used to construct ideal models considering no possible failures during execution and other specifics. Block 1.4.6 (Service Problem Management) at the third level is divided into seven groups; see Figure 3. In particular, it includes block 1.4.6.3 (Correct and Resolve Service Problem).

Block 1.4.6.3 (Correct and Resolve Service Problem) at the fourth level is divided into seven processes as follows (Figure 3):

- 1.4.6.3.1 Reassign/Reconfigure Failed Service;
- 1.4.6.3.2 Manage Service Restoration;
- 1.4.6.3.3 Implement Service Problem Work Arounds;
- 1.4.6.3.4 Invoke Support Service Problem Management Processes;
- 1.4.6.3.5 Review Major Service Problem;
- 1.4.6.3.6 Monitor Service Alarms Events;
- 1.4.6.3.7 Categorize Service Alarm Event.

The specification elements obtained at the fourth level can be used to construct a detailed model of a business process for further automation of operational management; see Figure 4.

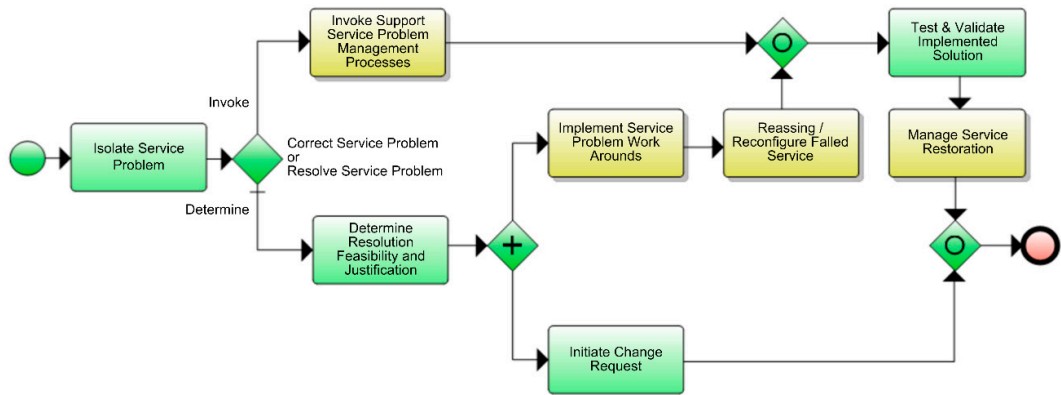

**Figure 4.** Flowchart for BP 1.4.6.3 (Correct and Resolve Service Problem) [9].

The described management system design with a set of modules can be implemented using the microservice architecture principles, which are used to create program systems composed of numerous multiple services interacting with each other. These modular components are intended for independent development, testing and deployment, which facilitates the creation of new services or a deep update of the existing ones if necessary. Other advantages of such modules are cost reduction, speed increase and customer service improvement. The microservice architecture guarantees independent scalability, a faster market entry for new services and also a higher efficiency of management.

### 3.2. Model of Distributed Management System for Next-Generation Networks

A distributed management system (DMS) for infocommunication networks can be expressed as a set of modules or program components (PCs) executed by processor modules (PMs). Each PC implements a corresponding business component (BC) as a microservice described within Frameworx. PMs can be implemented in the form of physical devices or virtual resources. PCs interact with each other through the network adapters of PMs using the capabilities of real or virtual networks. Any service in this management system is implemented by assigning a certain set of PCs located on separate PMs for the execution of a given BP. All services are created by uniting heterogeneous PCs that are formed using the virtual resources of several virtual networks. Instead of a local assignment to a specific PC, each service is implemented with a global distribution over the whole network.

Clearly, the performance of a distributed management system (DMS) depends on the quality of solving several tasks as follows:

- the distribution of PCs among PMs (note that (a) the number of PCs sets is often much greater than the number of PMs and (b) the real sequence of control transfers between PCs is approximated by an absorbing Markov chain);
- the prioritization of BPs with parallel execution as well as the elimination of dead states and interlocks during execution;

- the reduction of system cost to unite separate components of BPs, achieved via a rational integration of PCs.

A successful solution of these tasks yields an optimal DMS in terms of the TM Forum criterion (1)

$$\Phi = \min \sum_{k=1}^{L} P_k \times t_k(\theta, S, Q), \tag{1}$$

with the following notations: $t_k$ as the execution time of the $k$th BP; $\theta$ as a distribution method of PCs among PMs; $S$ as a service schedule of requests; $Q$ as an integration method for the separate solutions of PMs; $L$ as the number of BPs served by the system; finally, $P_k$ as the execution priority of the $k$th BP.

To distribute PCs over PMs, consider $n$ PCs $f_1, \dots, f_n$ and also $d$ PMs. Assume each two PCs, $f_i$ and $f_j$, are exchanging joint requests with a known frequency $P(i, j)$. The mean number of control transfers between the $i$th and $j$th PCs can be obtained using the measurements of a program monitor of the system. Find an analytical expression of the mean number of control transfers between PCs in the course of service implementation.

To this end, decompose the set of PCs into $d$ groups $\Phi = \{F_1, F_2, \dots, F_d\}$ so that

$$\bigcup_{i=1}^{n} f_i = \bigcup_{i=1}^{d} F_i \quad \text{and} \quad F_k \cap F_l = \varnothing, \quad k \neq l. \tag{2}$$

The frequency of conflicts on the $k$th PM is given by

$$C_k = \sum_{i, j} p(i, j). \tag{3}$$

Then the total frequency of conflicts can be calculated as

$$C = \sum_{k=1}^{d} C_k = \sum_{k=1}^{d} \sum_{i, j} p(i, j). \tag{4}$$

If PC$_i$ performs a transition between groups $F_s$ and $F_t$, then the optimality criterion has the variation (5)

$$\Delta C = \sum_{i \in PM_i} (P(i, j) + P(j, i)) - \sum_{i \in PM_s} (P(i, j) + P(j, i)). \tag{5}$$

All PC$_i$, $i = 1, \dots, n$, are decomposed into $d$ groups using a system of operators $R = \{R_{i\,t}, i = 1, \dots, n; t = 1, \dots, d\}$ that formalizes the transition of PC$_i$ to another class $t$, i.e., the operation $R_{i\,t}(\Phi)$. As a result, the optimal decomposition criterion has the increment $\Delta_{i\,t}(\Phi) = C(\Phi) - C(R_{i\,t}(\Phi))$, where $C(R_{i\,t}(\Phi))$ is the frequency of conflicts for the operation $R_{i\,t}(\Phi)$ and $C(\Phi)$ is the preceding frequency of conflicts.

Denote by $\Delta_{i\,t}^{i\,q}(\Phi)$ the increment of the values $\Delta_{i\,t}(\Phi)$ under transition to a new decomposition $R_{j\,q}\Phi$, where $R_{j\,q} \in R$, i.e.,

$$\Delta_{i\,t}^{i\,q}(\Phi) = \Delta_{i\,t}(R_{i\,q}\Phi) - \Delta_{i\,t}(\Phi), \quad i = 1, \dots, n, \quad q = 1, \dots, d. \tag{6}$$

Then

$$\Delta_{i\,t}(R_{j\,q}\Phi) = \Delta_{i\,t}(\Phi) + \Delta_{i\,t}^{j\,q}(\Phi); \quad \Delta_{i\,t}^{j\,q}(\Phi) = [P(i, j) + P(j, i)](\delta_{t\,q} + \delta_{s\,u} + \delta_{s\,q} - \delta_{t\,u}), \tag{7}$$

where $s$ and $u$ are the indexes of groups for the PCs $f_i$ and $f_j$ in the initial decomposition; $t$ and $q$ are the indexes of their groups in the new decomposition; finally, $\delta_{t\,q} = \begin{cases} 1 \text{ if } t = q; \\ 0 \text{ if } t \neq q. \end{cases}$

For this criterion, the sequential improvement algorithm has the form (8)

$$\Delta_{i\,t}^{i\,q}(\Phi) = \Delta_{i\,t}\big(R_{i\,q}\Phi\big) - \Delta_{i\,t}(\Phi).\tag{8}$$

The algorithm guarantees a rational distribution of PCs among PMs under the assumption that the sequence of control transfers between PCs is described by a Markov chain.

The block diagram of this distribution algorithm is presented in Figure 5.

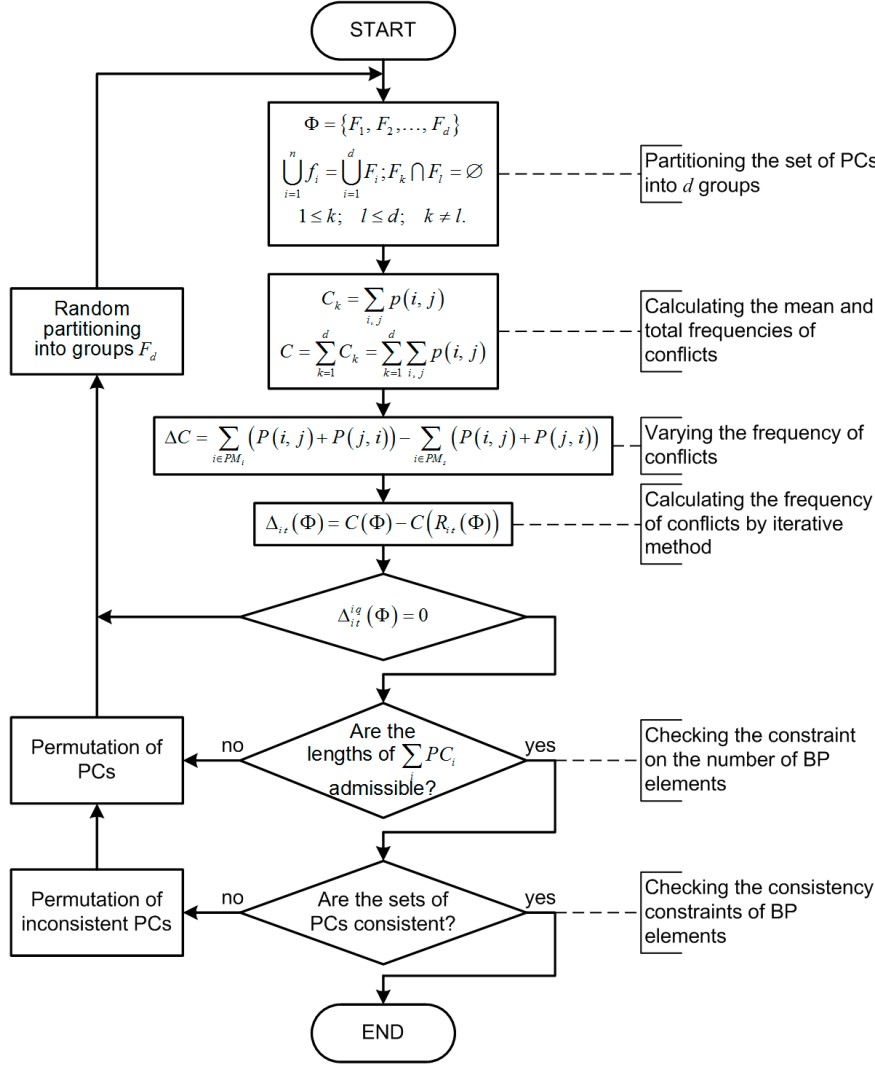

**Figure 5.** Block diagram of program components distribution algorithm.

In accordance with this algorithm, the quality of distribution of PCs among PMs is assessed by the frequency of resource conflicts. The variation $\Delta_{i\,t}^{i\,q}(\Phi) = \Delta_{i\,t}\big(R_{i\,q}\Phi\big) - \Delta_{i\,t}(\Phi)$ of this frequency is calculated if a PC performs a transition between groups. In a finite number of iterations, the algorithm converges to a local minimum.

### 3.3. Elimination of Dead States and Interlocks

DMSs for infocommunication networks have complex operation algorithms with parallel processes. Note that some processes are interdependent because they share the same resources (e.g., hardware components, software tools, current information). Due to resource sharing, the interaction of parallel

processes has to be properly organized: their independent execution may cause errors, dead states or interlocks [10–12].

The dead states and interlocks occurring in parallel business processes are eliminated by scheduling. The efficiency of elimination is assessed using the indicator (9)

$$T_{\min}(S) = \min\{t_i(S)\}, \tag{9}$$

where $S$ denotes a schedule and $t_i$ $(S)$ is the service time of the $i$th request, subject to the constrain (10)

$$t^j_{p+1} - t^i_p \geq Z^i_L - Z^j_L + t_{jL}, \; L = \overline{1, \, k}. \tag{10}$$

The notations are the following: $p$ as the index of the request of type $i$; $(p+1)$ as the index of the request of type $j$; $L$ as the serial number of the PM; $Z^i_L$ and $Z^j_L$ as the time cost to execute the requests of types $i$ and $j$ on the $L$th PM; finally, $t_{jL}$ as the time to execute the request of type $j$ on the $L$th PM. Let $l_{i\,j} = \max\{Z^i_L - Z^j_L + t_{jL}\}$. Then the optimal service schedule of all requests is defined by the condition $t^j_{p+1} - t^i_p = l_{i\,j}$. Describe the service procedure of all $n = \sum\limits_{k=1}^{r} n_k$ requests using a directed symmetric graph $(X, U)$, where $X = \{0, \, 1, \dots, \, r\}$ and $U$ is the set of arcs $(i, j)$, $0 \leq i, \; j \leq r$, each associated with the value $l_{ij}$. In this case, the optimal service schedule of all requests is the smallest loop in the graph that passes $n_k$ times through each vertex.

If $x_{i\,j}$ is the number of arcs in the desired loop, then

$$\sum\limits_{i, \, j=0}^{r} l_{i\,j} x_{i\,j} \to \min, \; \sum\limits_{j=0}^{r} x_{i\,j} = \sum\limits_{j=0}^{r} x_{j\,i} = n_i, \; i = \overline{0, \, r}, \; x_{i\,j} \geq 0, \; i, \; j = \overline{0, \, r}. \tag{11}$$

The interlocks of parallel business processes are eliminated using an interpretation of the classical file sharing example with two processes [11–14] and also the methodology of colored Petri nets (CPN) [15], with implementation in CPN Tools [16,17].

Consider parallel BPs consisting of a sequence of operations $t_1, \dots, t_k$ performed by PCs so that each operation corresponds to a transition in a Petri net; see Figure 6. The CPN Tools user interface determines a marking of a Petri net. Markers (often called tokens) contained in certain positions are highlighted in green color, with specification of their number and time delay (e.g., 1'1@0). Additionally, a green color is used for the transitions that can be activated at a current time.

A certain number of asynchronous parallel processes are competing for the right of resource use (RES). When a process is holding a resource, the sequence of its operations is being performed and the resource is considered to be busy. Since the same resource can be required for several processes, there may exist dead states and interlocks for them, as is demonstrated in Figure 7. Clearly, at the current time the network has no transitions that can be activated.

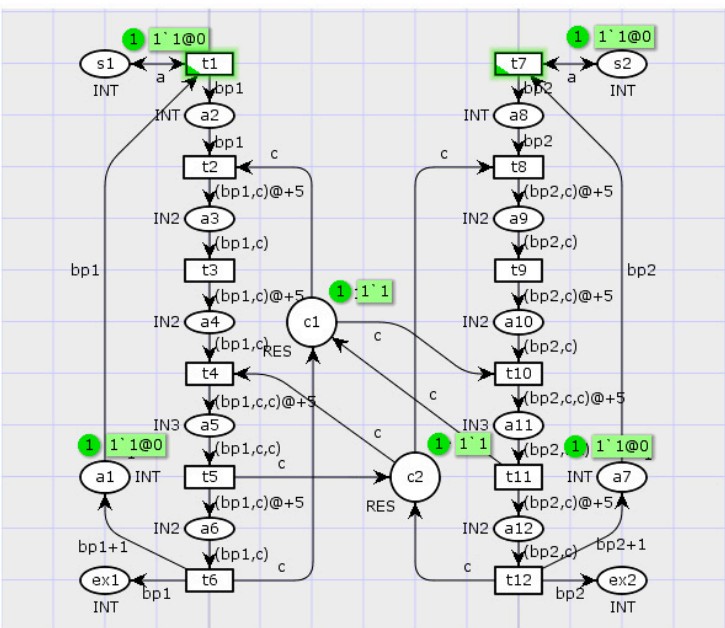

**Figure 6.** Initial state of execution model for two parallel BPs.

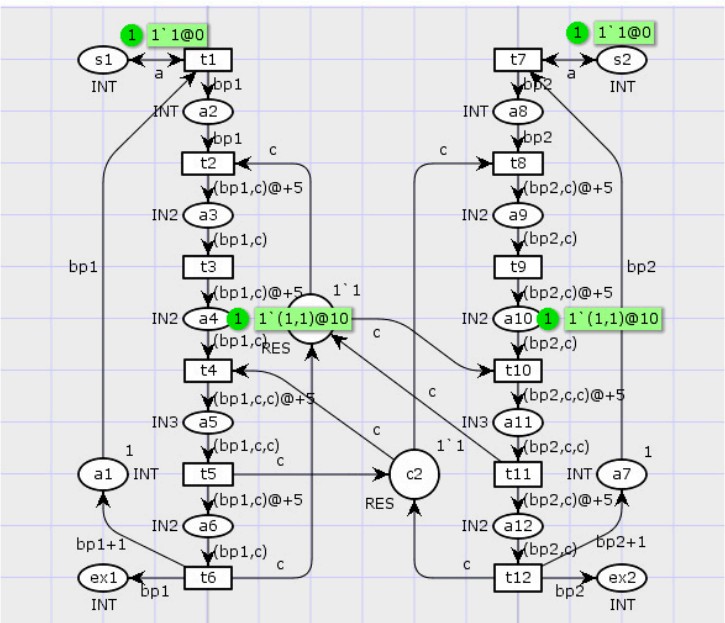

**Figure 7.** Execution model for two parallel BPs in interlock state.

The model will be analyzed using the reachable marking graph of the Petri net (Figure 8). The states of a Petri net from which all paths of the reachable marking graph lead to a dead state (in this example, $S_{33}$) are called pre-dead states (in this example, $S_{22}$, $S_{23}$ and $S_{32}$). A set composed of the dead and pre-dead states is called the set of forbidden states [10]. Obviously, for a faster execution of processes, all conflicts must be eliminated using the available system parallelism as much as possible. Consider some ways to eliminate process interlocks and dead states.

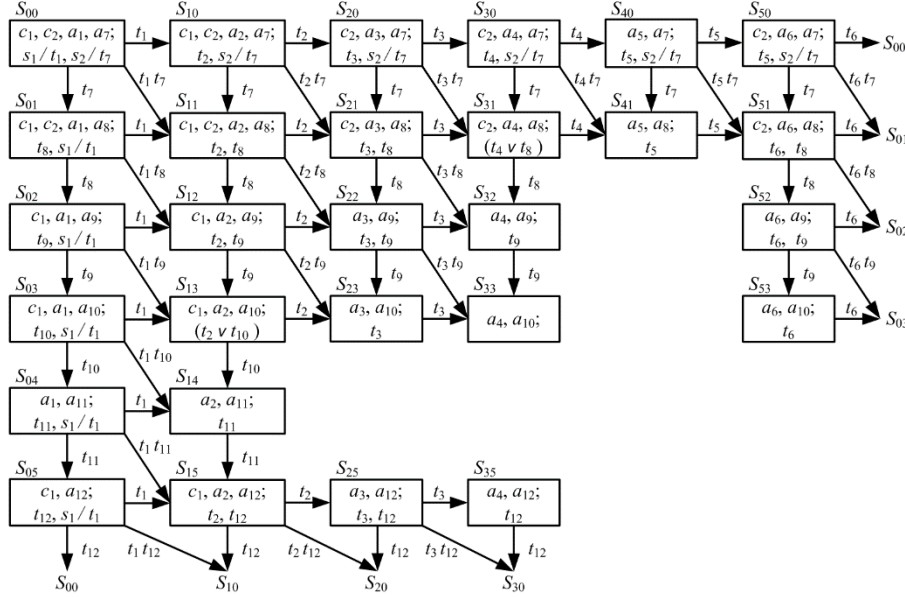

**Figure 8.** Reachable marking graph of original Petri net.

The first approach is to use a well-timed forced locking of processes [10]. To this effect, define the minimal number of processes to be locked and also the minimal number of states to preserve this interlock. Then transform the reachable marking graph by removing all the edges that connect dangerous and forbidden states. This procedure yields a graph containing the safe states only.

In this example, the state $S_{11}$ is the root of two dangerous segments, $S_{21} - S_{31}$ and $S_{12} - S_{13}$ (Figure 8). The well-timed forced locking of undesired processes can be implemented by introducing an input position $c_b$ for the transitions $t_2$ and $t_8$ in the Petri net (Figure 9). In this case, following the activation of the transition $t_2$ ($t_8$) and further evolvement of the process $bp_1$ ($bp_2$, respectively), the token is removed from the position $c_b$ and the process $bp_2$ ($bp_1$, respectively) is locked.

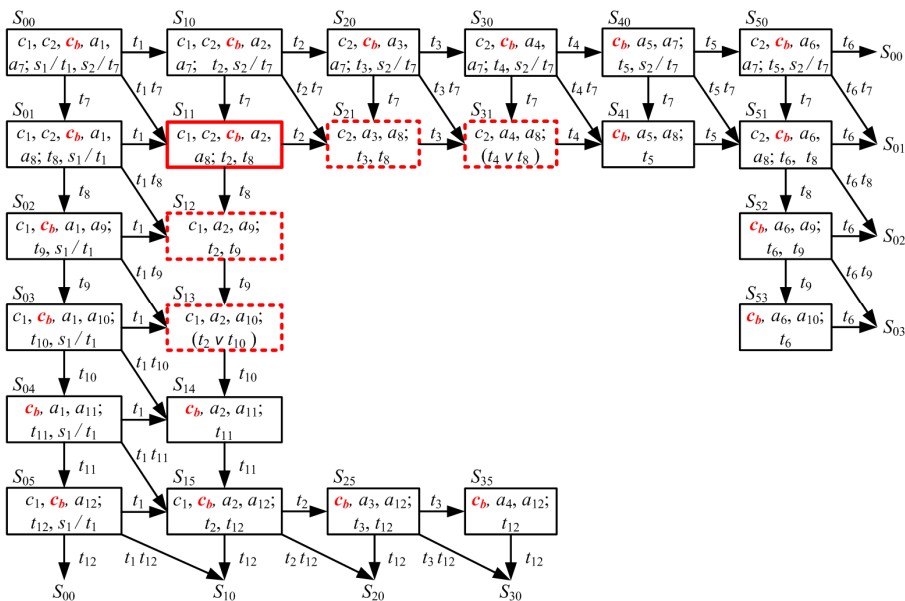

**Figure 9.** Transformed reachable marking graph of Petri net.

In accordance with the transformed reachable marking graph (Figure 9), the state $S_{31}$ ($S_{13}$) is the last state of the chosen dangerous segment. Hence, the lock can be lifted after the activation of

the transition $t_4$ ($t_{10}$, respectively). To this end, the position $c_b$ must be the output position for the transitions $t_4$ and $t_{10}$, as is illustrated in Figure 10.

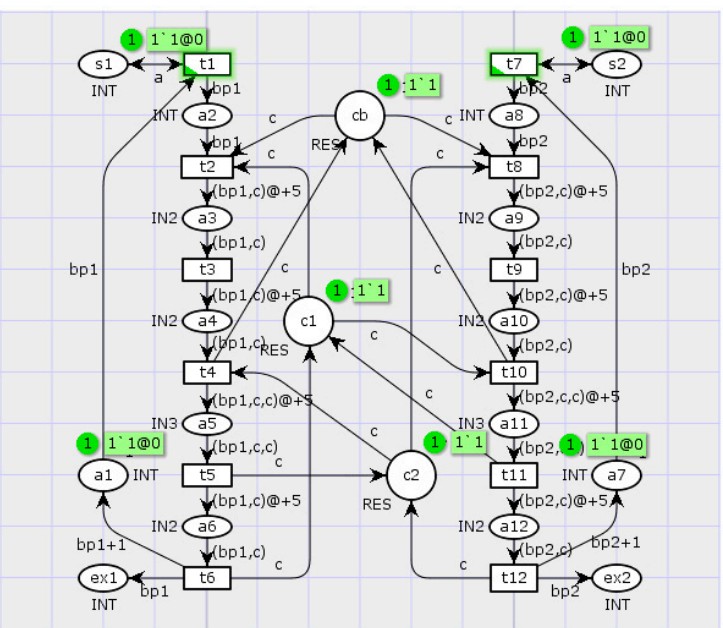

**Figure 10.** Initial state of execution model for two parallel BPs with well-timed forced locking.

The transformed reachable marking graph (Figure 9) contains the position $c_b$ in all states except for the ones corresponding to dangerous segments. The states $S_{12}$, $S_{13}$, $S_{21}$ and $S_{31}$, which are dangerous in the original graph, become safe lock states; the state $S_{11}$ becomes the conflict state. In addition, the forbidden states $S_{22}$, $S_{32}$, $S_{23}$ and $S_{33}$ turn out to be unreachable in the transformed graph. The simulation of the transformed Petri net over 100 steps have testified to the efficiency of this well-timed forced locking procedure; see the simulation results in Figure 11.

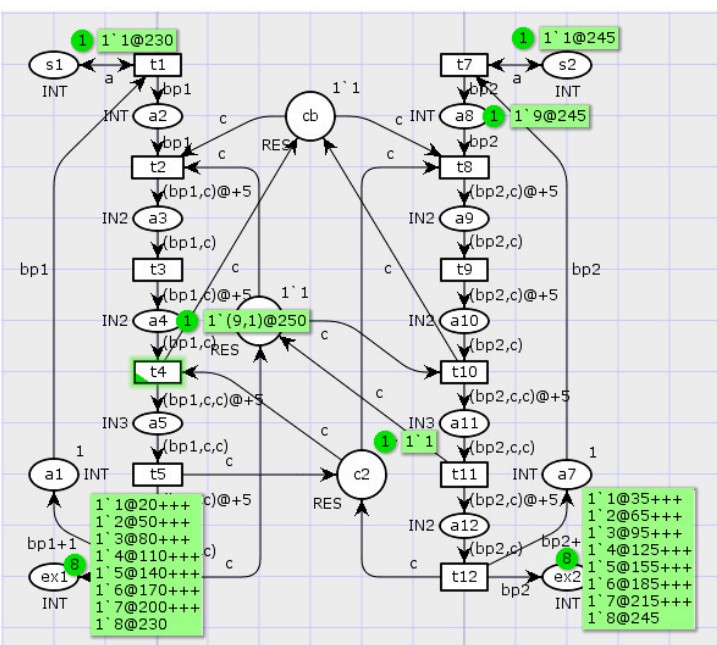

**Figure 11.** Transformed execution model for two parallel BPs: simulation results.

The second approach to eliminate the dead states and interlocks of processes is to allocate the additional resources required for their simultaneous execution (Figure 12). In this case, the positions $c_1$ and $c_2$ have single tokens, which corresponds to two units of homogeneous resource.

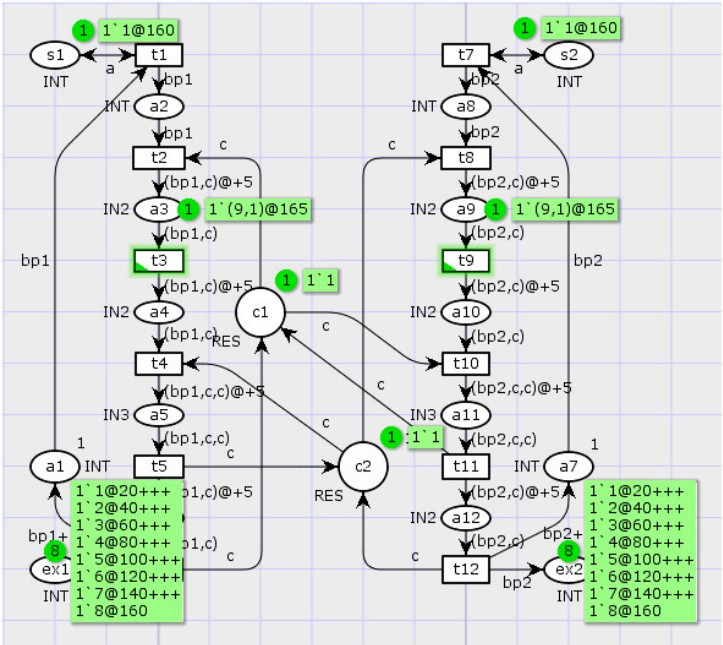

**Figure 12.** Execution model for two parallel BPs with allocation of additional resources: simulation results.

Next, the third approach to eliminate the dead states and interlocks of processes is to capture simultaneously all the resources required for a process (Figure 13). For lifting the interlock of the first process, a position $c_{21}$ is added to the original network that holds the second resource.

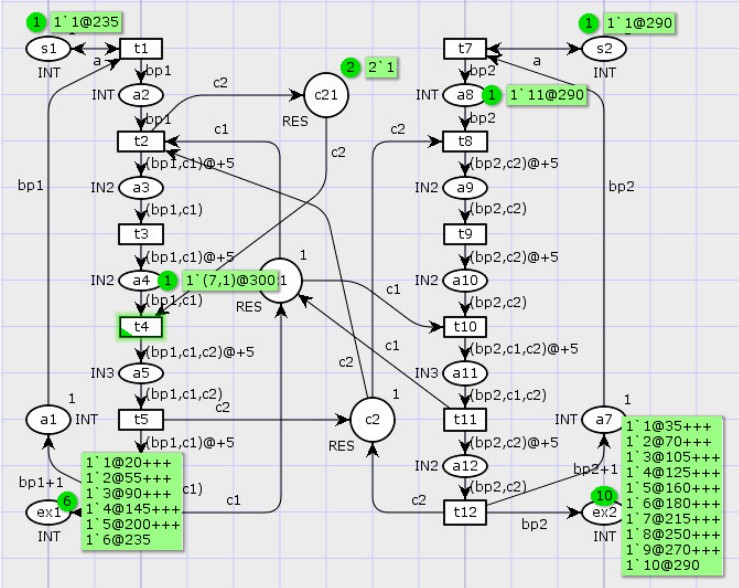

**Figure 13.** Execution model for two parallel BPs with monopolistic capture of resources: simulation results.

The last (fourth) approach to eliminate the dead states and interlocks is to arrange the capture of resources. Serial numbers are assigned for all types of resources and a capture discipline is defined for all processes. In the transformed model, this approach is implemented by reassigning serial numbers for resources and specifying choice rules for the transitions $t_2$, $t_4$, and $t_8$, $t_{10}$ (Figure 14).

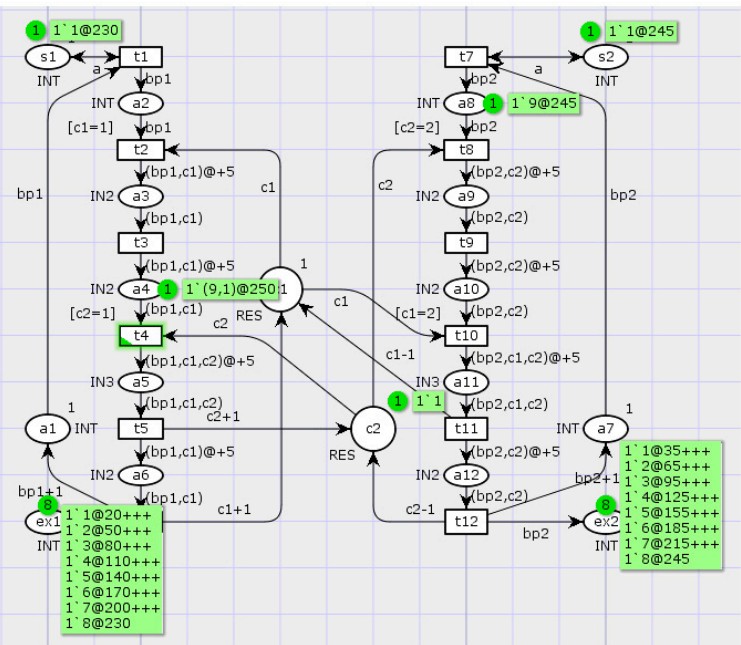

**Figure 14.** Execution model for two parallel BPs with arranged capture of resources: simulation results.

The developed models eliminate the interlocks of parallel processes. The methodology of colored Petri nets is used to analyze the complete state space of the model in order to improve the reliability of the computing system and also to satisfy the requirements. The suggested methods and software solutions allow us to accelerate and simplify program development. They are finely integrated into the standard software approaches and methods and ready to be applied in practice.

### 3.4. Analytical Model of Integration System for PCs

In accordance with the above results, a simultaneous execution of several logically independent parallel processes on the same resource actually increase the system cost of operational management and reduce the system performance; for a considerable number of processes, it may even cause network resource deficit. The structure of software-defined networking (SDN) and, in particular, the approaches to organize logically centralized control of network elements were described in the ITU-T Recommendations, Y.3300 series [18]; also see Figure 15. The application-control interface is intended to implement program control of abstract network resources. The resource-control interface is intended to implement the functions of logically centralized control of network resources.

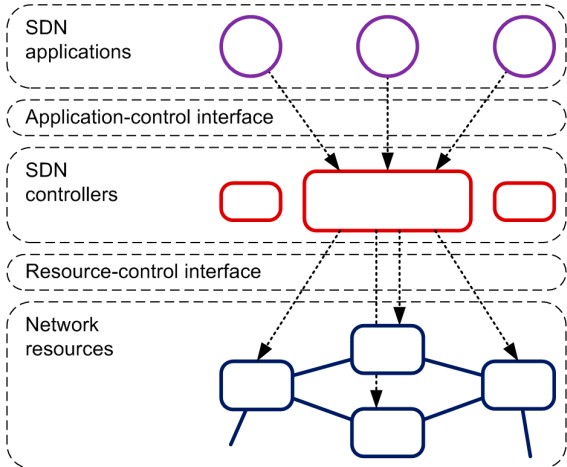

**Figure 15.** Structure of software-defined networking for centralized control of network resources [18].

Assume the transitions of requests between the elements of network resources as well as their withdrawals from the network are described by an irreducible Markov chain; in addition, let the processes occurring in the network be the multidimensional birth and death processes. Then, an SDN controller can be represented as a root node $M$ of the network while the network nodes—the program components of a physically distributed application—act as the other ($M$–1) nodes that execute a corresponding business process. For the requests of the class $R = (1, r)$, the transition probabilities can be described by a matrix (12)

$$\|P_r\left(\overline{n_i^j}\right)\| = \begin{vmatrix} P_{r,1,1} & P_{r,1,2} & \ldots & P_{r,1,M} \\ P_{r,2,1} & P_{r,2,2} & \ldots & P_{r,2,M} \\ \ldots & \ldots & \ldots & \ldots \\ P_{r,M,1} & P_{r,M,2} & \ldots & P_{r,M,M} \end{vmatrix}, \tag{12}$$

where $P_r\left(\overline{n_i^j}\right)$ denote the probabilities of transition and $\overline{n_i^j}$ is the number of requests passing from node $i$ to node $j$.

Write the multidimensional random interaction process of PCs as (13)

$$N(t) = \{n_1(t), n_2(t), \ldots, n_R(t)\}; \tag{13}$$

the probability that $k$ requests of the $r$th class have to be served as $P(k) = P_1(k_1) \cdot P_2(k_2) \cdot \ldots P_n(k_n)$, where $k = (k_1, k_2, \ldots, k_n)$ and $k_i = (k_{i1}, k_{i2}, \ldots, k_{ir})$.

Such a network can be described in the multiplicative form [19]

$$P(n) = G^{-1}(N_1, N_k, \ldots, N_R) \prod_{i=1}^{M} Z_i(n_i), \tag{14}$$

where $P(n)$ denotes the probability that the network is in state $n$,

$$n = (n_1, n_{2.}, \ldots, n_M); \quad G(N_R) = \sum_{k} \prod_{i=2}^{N} \left(\frac{\mu_1 P_i}{\mu_i}\right)^{k_i};$$

$$Z_i(n_i) = \frac{n_i!}{\prod\limits_{R=1}^{r} \mu_i(R)} \prod_{r=1}^{R} \frac{1}{n_{ir}!} l_{ir}^{n_{ir}},$$

$$l_{ir} = \sum_{i=1}^{M} l_{ir} P_{ij}(r), i = \overline{1, M}, r = \overline{1, R}.$$

As a result,

- the SDN controller capacity for a request of the $r$th class is (15)

$$\lambda_{ir}(N_R) = \sum_{n_k=1}^{N_R} P_i(n_R, N_r) \frac{n_{ir}}{n_i} \mu_i(n_i);$$

(15)

- the number of served requests of the $r$th class is (16)

$$L_{ir}(N_R) = \sum_{n_R=1}^{N_R} P_i(n_R, N_R) n_r;$$

(16)

- the mean waiting time for a request of the $r$th class is (17)

$$T_{ir}(N_r) = \frac{[1 + L_i(N_R - 1)]}{M_{ir}}.$$

(17)

In practice, the values of these indicators depend on system load at request arrival times. In some state the system may block a successive call. In this case, the call is repeated $N$ times till being served or rejected; see Figure 16.

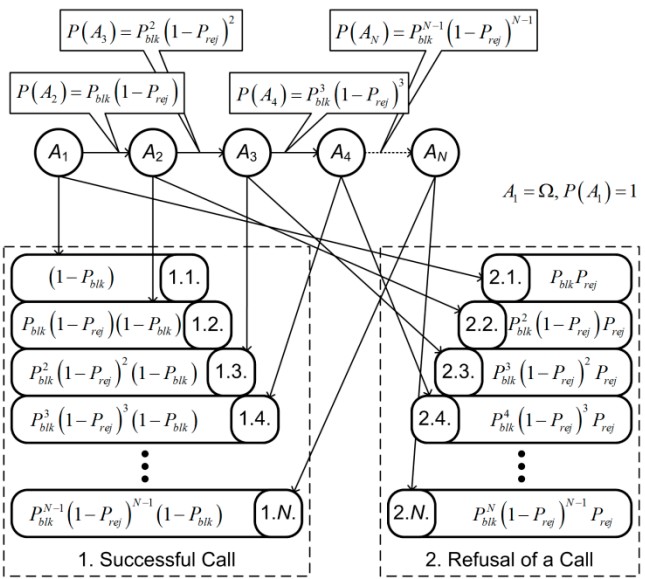

**Figure 16.** Diagram of request service procedure.

Hence, the probability of a successful call is (18)

$$P_{suc}(N) = P_{blk}^{N-1}\left(1 - P_{rej}(1)\right)\left(1 - P_{rej}(2)\right)\ldots\left(1 - P_{rej}(N-1)\right)(1 - P_{blk}).$$

(18)

The probability of a rejected call (19)

$$P_{rej}(N) = P_{blk}^{N}\left(1 - P_{rej}(1)\right)\left(1 - P_{rej}(2)\right)\left(1 - P_{rej}(N-1)\right).$$

(19)

Some trivial transformations yield (20), (21)

$$P_{suc}(N) = (1 - P_{blk})\left\{1 + \sum_{i=1}^{N-1} P_{blk}^i \prod_{j=1}^{i} (1 - P_{rej}(j))\right\}, \tag{20}$$

$$P_{rej}(N) = P_{blk}\left\{P_{rej}(1) + \sum_{i=1}^{N-1}\left[P_{blk}^i P_{rej}(i+1) \prod_{j=1}^{i} (1 - P_{rej}(j))\right]\right\}. \tag{21}$$

For a maximum speed of 10 million requests (request flows) per second and a maximum delay of 50 µs, the system has to serve up to 50 million flows per second. This example effectively illustrates that the main load on rather limited computational resources of a network multicore controller may considerably affect its performance. The incoming requests for such resources from several network elements simultaneously may cause network blocking. This problem is solved using the distributed control of parallel processes: all PCs are divided into segments associated with corresponding controllers. Note that each segment must contain the PCs with the greatest probabilities of interaction; see Figure 17.

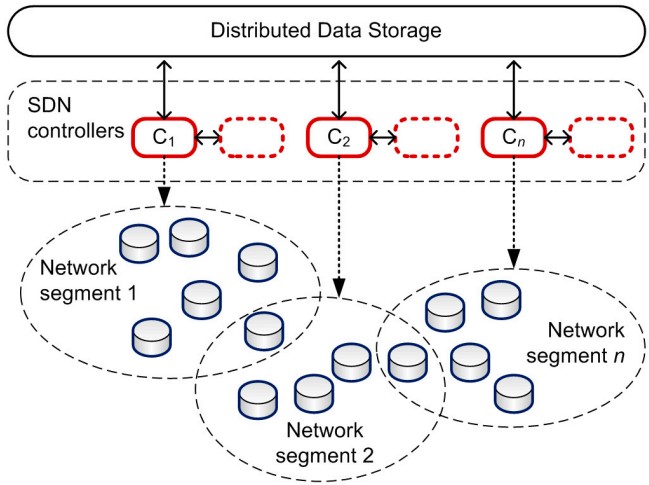

**Figure 17.** Diagram of distributed control of network elements.

A segmentation $\Phi$ of the set of PCs is described by matrices (22)

$$V = \|V_{ik}\|_{i=1,\,k=1}^{n,\,m} \text{ and } L = \|l_{ij}\|_{i,j=1}^{n}, \tag{22}$$

where $V_{ik} = \begin{cases} 1 \text{ if } BC_i \in \Phi_k, \\ 0 \quad \text{otherwise}; \end{cases}$ $l_{ij} = \begin{cases} 1 \text{ if } BC_i \in \Phi_k; \\ 0 \text{ otherwise}. \end{cases}$

Obviously, the elements of the matrix $L$ are expressed through the elements of the matrix $V$ as follows: $l_{ij} = \sum_{k=1}^{n} V_{ik} V_{jk0}$.

Let $m_i$ be the mean number of executions of the $i$th PC. Control is transferred to the $j$th block with a probability $p_{ij}$. Hence, the product $m_i P_{ij}$ gives the mean number of control transfers between the $i$th and $j$th blocks.

The mean number of transitions from the $k$th fragment can be written as (23)

$$\sum_{k=1}^{n} \sum_{j=1}^{n} m_i p_{ij} V_{ik}(1 - V_{jk}), \tag{23}$$

where $V_{jk}$ takes into account the BCs from the $k$th fragment only and the factor ($1$-$V_{jk}$) eliminates the transitions from the blocks $j \neq i$ belonging to the $k$th segment. Then the mean number of intersegment transitions is (24)

$$C = \sum_{k=1}^{n}\sum_{i=1}^{n}\sum_{j=1}^{n} m_i p_{ij} V_{ik}\left(1 - V_{jk}\right)_0. \tag{24}$$

Therefore, the expression $\sum_{k=1}^{n}\sum_{i=1}^{n}\sum_{j=1}^{n} m_i p_{ij} V_{ik} V_{jk}$ determines the mean number of transitions between the BCs of a segment. Then the optimization problem has the form $\sum_{k=1}^{n}\sum_{i=1}^{n}\sum_{j=1}^{n} m_i p_{ij} V_{ik} V_{jk} = \sum_i \sum_j q_{ij} l_{ij} \to$ min subject to the constraint $\sum_{i \in \Phi_k} S_i \leq B$, where $S_i$ is the memory size required for executing the $j$th PC; B denotes a maximum admissible memory size for a segment.

Construct the transition probability matrix for a Markov chain of $n$ PCs using the formulas $Q = \|P_{ij}\|_{i,j=1}^{n}$, $P_{ij} == \frac{g_{ij}}{\sum_j g_{ij}}$, where $g_{ij}$ is the mean number of control transfers between the $i$th and $j$th PCs and $P_{ij}$ is the probability of control transfer to the $j$th PC.

Describe the segmentation $\Phi$ of the set of all PCs in the following way:

$$V = \|v_{ik}\|_{i=1,\ k=1}^{n,\ m}, \quad L = \|l_{ij}\|_{i,j}^{n}, \quad \text{where } v_{ik} = \begin{cases} 1 \text{ if } PC_i \in \Phi_k, \\ 0 \text{ otherwise}; \end{cases} \quad l_{ij} = \begin{cases} 1 \text{ if blocks } i, j \in \Phi_k, \\ 0 \text{ otherwise}. \end{cases}$$

Write the constraints $\sum_{j=1}^{n} S_j l_{ij} \leq B$, where $S_i$ is the memory size for the $i$th PC; B is the memory size for a segment $\Phi_k$.

Calculate the mean number of transitions from the $k$th segment as $C = \sum_{i=1}^{n}\sum_{j=1}^{n} m_i P_{ij} v_{ik}$, where $m_i$ is the mean number of executions of the $i$th PC.

Minimize the mean number of intersegment transitions $C = \sum_{k=1}^{n}\sum_{i=1}^{n}\sum_{j=1}^{n} m_i P_{ij} v_{ik} v_{jk} = \sum_{i=1}^{n}\sum_{j=1}^{n} g_{ij} l_{ij}$, where $P_{ij} = \frac{g_{ij}}{\sum_j g_{ij}}$, subject to the constraints $\sum_{j=1}^{n} S_j l_{ij} \leq B$, $l_{ij} + l_{it} + l_{jt} - 2 l_{ij} l_{it} l_{jt} \leq 1$ for $\forall i, j, t$. These constraints guarantee that each PC belongs to a single segment $\Phi_k$.

### 3.5. Simulation Results of Request Service Procedure

An example of the graph of the request implementation procedure is shown in Figure 18. The memory distribution among the elements of a BP (the so-called business components, BCs) is presented in Table 1.

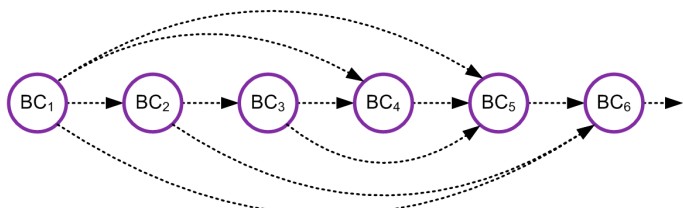

**Figure 18.** Graph of request implementation procedure.

**Table 1.** Memory distribution among business components (BCs).

| | Elements of Business Process | | | | | |
|---|---|---|---|---|---|---|
| | 1 | 2 | 3 | 4 | 5 | 6 |
| Size of BC, in Kb | 3000 | 2000 | 1500 | 1000 | 700 | 500 |

The mean number of control transfers $g_{ij}$ can be obtained using a program monitor of this system. The results of measurements are combined in Table 2.

**Table 2.** The mean number of control transfers $g_{ij}$.

| Value $i$ | Value $j$ | | | | | |
|:---:|:---:|:---:|:---:|:---:|:---:|:---:|
| | **1** | **2** | **3** | **4** | **5** | **6** |
| 1 | 0 | 372 | – | 12 | – | – |
| 2 | – | 0 | 270 | – | – | 9 |
| 3 | – | – | 0 | 227 | 13 | – |
| 4 | – | – | – | 0 | 732 | – |
| 5 | – | – | – | – | 0 | 21 |
| 6 | – | – | – | – | – | 0 |

Here the desired variables are the elements of the matrix $L = \|l_{ij}\|_{i,j=1}^{n}$, where $l_{ij} =$
$\begin{cases} 1 \text{ if blocks } i \text{ and } j \text{ belong to a single fragment,} \\ 0 \text{ otherwise.} \end{cases}$

The implementation times of the sequence of PCs are given in Table 3.

**Table 3.** Implementations of different configurations of BP elements.

| The Number of Intersegment Transitions | The Elements of Matrix $L$ | | | | | | | | Implementation Time, in ms |
|:---:|:---:|:---:|:---:|:---:|:---:|:---:|:---:|:---:|:---:|
| | $l_{12}$ | $l_{13}$ | $l_{14}$ | $l_{23}$ | $l_{25}$ | $l_{26}$ | $l_{56}$ | $l_{46}$ | |
| 0 | 1 | 1 | 1 | 1 | 1 | 1 | 1 | 1 | 3.57 |
| 30 | 0 | 0 | 1 | 1 | 1 | 1 | 1 | 1 | 3.72 |
| 100 | 1 | 1 | 0 | 0 | 1 | 1 | 1 | 1 | 3.85 |
| 200 | 1 | 1 | 1 | 1 | 0 | 0 | 1 | 1 | 4.03 |
| 1000 | 1 | 1 | 1 | 1 | 1 | 1 | 0 | 0 | 5.71 |
| 2000 | 0 | 0 | 0 | 0 | 1 | 1 | 1 | 1 | 7.31 |

Clearly, the optimal combination consists of the following PCs: 1–5, 1–6, 1–4, 2–3, 2–5, 2–6, 4–6, 5–6. All PCs of a business process have to be divided into two segments: PCs 1–4–5–6 and 2–3, or PCs 1–4 and 2–3–5–6. The execution time of the business process (service implementation) has been reduced from 4.416 to 3.681 ms, which is 17%.

The main limitations of the developed models, which are related to the accuracy of the research results, are determined by the assumption that the phases of receiving and processing packages are independent.

## 4. Conclusions

In this paper, the formal design problem of distributed operational management and support systems for communication companies and the main approaches to this problem have been considered. An iterative convergent algorithm for distributing program components among program modules that implements a corresponding microservice or business component in the Frameworx description has been presented. A service schedule of requests with the colored Petri nets formalization that minimizes the number of delayed requests has been developed. An integration algorithm for program components that minimizes the system cost of their interaction has been proposed. All these results contribute to the efficient design of distributed operational management systems for next-generation telecommunication networks and also determine a flexible, software-defined and cost-saving architecture of management systems for infocommunication networks and services.

**Author Contributions:** Formal analysis, S.Y.; Methodology, V.M.; Validation, G.L.; Visualization, N.B.

**Funding:** This research was funded by the Russian Foundation for Basic Research (RFBR), grant number 19-07-00856\19.

**Conflicts of Interest:** The authors declare no conflict of interest.

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
