# Peer review of "Distributed Management Systems for Infocommunication Networks: A Model Based on TM Forum Frameworx"

_computers, doi:10.3390/computers8020045_

Round 1
Reviewer 1 Report
This paper presents a model for distributed operational management systems. The proposed model is novel and interesting. The paper is well written, except that there are a few typos. For example, "Frameworx" in the paper title should be "Framework".
Author Response
Good day. Thank You so much for the recommendations. We send the article file with a note about the changes. Thank You.
Reviewer 2 Report
The article looks good. However, the authors can perhaps add a section on limitations of the experiment. Also figure 4 needs improvement. the words in one of the boxes was not clear.
1) The very first sentence in the introduction section needs improvement due to wrong use of lists or commas. I would therefore advise improving English
2) The section 2 on results can be replace by objectives. Results must come at the end
3) It is better to have all equations numbered and referenced in the body
4) Just before the conclusion section the authors can write a section on limitations of the experiment or thesis
Author Response
Good day.
Thank you very much for the recommendations.
We send the article file with notes that highlight the changes made.
